# The Molecular Genetics of Dissociative Symptomatology: A Transdiagnostic Literature Review

**DOI:** 10.3390/genes13050843

**Published:** 2022-05-08

**Authors:** Ravi Philip Rajkumar

**Affiliations:** Department of Psychiatry, Jawaharlal Institute of Postgraduate Medical Education and Research (JIPMER), Puducherry 605 006, India; ravi.psych@gmail.com; Tel.: +91-413-229-6280

**Keywords:** dissociative disorder, depersonalization, derealization, dissociative identity disorder, dissociative fugue, dissociative amnesia, dissociative symptoms, genetics, genome-wide association study

## Abstract

Dissociative disorders are a common and frequently undiagnosed group of psychiatric disorders, characterized by disruptions in the normal integration of awareness, personality, emotion and behavior. The available evidence suggests that these disorders arise from an interaction between genetic vulnerability and stress, particularly traumatic stress, but the attention paid to the underlying genetic diatheses has been sparse. In this paper, the existing literature on the molecular genetics of dissociative disorders, as well as of clinically significant dissociative symptoms not reaching the threshold of a disorder, is reviewed comprehensively across clinical and non-clinical samples. Association studies suggest a link between dissociative symptoms and genes related to serotonergic, dopaminergic and peptidergic transmission, neural plasticity and cortisol receptor sensitivity, particularly following exposure to childhood trauma. Genome-wide association studies have identified loci of interest related to second messenger signaling and synaptic integration. Though these findings are inconsistent, they suggest biologically plausible mechanisms through which traumatic stress can lead to pathological dissociation. However, methodological concerns related to phenotype definition, study power, and correction for the confounding factors limit the value of these findings, and they require replication and extension in studies with better design.

## 1. Introduction

Dissociative disorders represent an important group of psychiatric disorders occurring in response to stress or trauma [1]. According to the American Psychiatric Association’s Diagnostic and Statistical Manual for Mental Disorders, Fifth Edition (DSM-5), these disorders are characterized by “a disruption of and/or discontinuity in the normal integration of consciousness, memory, identity, emotion, perception, body representation, motor control, and behavior” [2]. The disorders in this group include depersonalization/derealization disorder, dissociative amnesia, and dissociative identity disorder. The estimated lifetime prevalence of these disorders is around 10% in general population samples [3,4,5], with higher rates being reported in patients seeking treatment for other psychiatric disorders [6,7] and in groups exposed to trauma [8,9,10]. In addition to these discrete syndromes, clinically significant symptoms of dissociation occur in a wide range of other psychiatric disorders, including post-traumatic stress disorder (PTSD), personality disorders, somatic symptom disorders, depression, anxiety disorders, and eating disorders [11]. Dissociative disorders tend to run a chronic course [12], and are associated with elevated levels of disability, impaired quality of life, a high economic cost, and a significantly increased risk of suicide attempts [13,14]. In spite of the substantial burden associated with these conditions, patients suffering from them often go undiagnosed and untreated [7,15]. This is due to several factors, including variations in criteria across diagnostic systems [16], negative attitudes towards these disorders among both the general public and mental health professionals [17,18], and therapeutic pessimism arising from a paucity of evidence-based treatment approaches [19,20].

From a psychological perspective, research into the etiology of dissociative disorders has centered on “post-traumatic” and “socio-cognitive” models; however, more recent evidence suggests that neither model is completely satisfactory. Instead, these disorders should be understood as involving impairments in multiple domains of mental functioning, including self-regulation, reality testing, cognitive attributions, and higher-order cognitive processes [21]. Neurobiological evidence converges with this perspective, suggesting that dissociative disorders are best understood as belonging to a group of trauma-related disorders which share certain key structural and functional abnormalities. These include grey matter volume reductions in key limbic system structures, the dysregulation of prefrontal–limbic circuitry, and the altered functioning of the hypothalamic–pituitary-adrenal (HPA) axis [22]. These structural and functional alterations may underlie the multiple domains of impairment identified in psychological research. A recent meta-analysis identified reductions in the volume of hippocampus, basal ganglia and thalami, increased levels of the neuropeptide transmitters oxytocin and prolactin, and reduced levels of the inflammatory marker tumor necrosis factor alpha (TNF-α) as replicable biological markers of dissociative disorders [23]. Attempts to integrate the results of biological and psychological research into possible models of pathogenesis have already been made in the case of dissociative identity disorders [24] and dissociative amnesia [25,26].

A key finding from empirical research is that dissociation is a frequent, but not universal, response following exposure to a traumatic stressor. For example, around 20–25% of individuals exposed to a disaster may experience transient dissociative symptoms, but only a small proportion of them fulfill criteria for a later dissociative disorder [27]. Similarly, a study of women who had been held captive and subjected to sexual violence by a militant group found that only 41% met criteria for a dissociative disorder [28]. The finding that dissociation is not an invariable result of traumatic stress, even when severe and prolonged, suggests that some of this variation is due to an innate predisposition which is at least partly genetic in origin [29,30]. This hypothesis is supported by the results of twin studies which suggest that dissociation has a heritable component [31,32]. Understanding the contributions of specific genetic factors to dissociative disorders, whether in terms of monogenic, polygenic or epigenetic processes, would deepen our understanding of the stress–diathesis equation in these disorders. The insights obtained from molecular genetic data could potentially open the door to more effective forms of prevention and treatment, including early intervention strategies following a mass traumatic event. The current paper attempts to aid in this understanding by reviewing the scientific literature on the genetics of dissociative symptoms and disorders.

## 2. Materials and Methods

The current review was conducted in line with the principles outlined in the PRISMA guidelines [33]. A comprehensive search of the PubMed, Scopus and ProQuest literature databases was carried out using the key search terms “dissociative disorder”, “dissociative disorders”, “dissociative identity disorder”, “depersonalization”, “derealization”, and “dissociative amnesia” in association with “genetic”, “gene”, “polymorphism”, “genome”, “genome-wide”, “epigenetic”, “heritability”, and “inheritance”. All articles published up to 31 March 2022 were included. The complete search strategy is provided in Appendix A.

Papers were included in the review if they presented the results of genetic studies of either (a) dissociative disorders, as defined by DSM-5, or (b) clinically significant dissociative symptoms (“pathological dissociation”) either alone or in conjunction with other psychiatric disorders, as measured using a standardized rating scale. A total of 588 citations were retrieved through the above search. After the removal of 186 duplicate citations, 402 citations (title and abstract) were screened. At this stage, 255 citations unrelated to the subject were excluded, and 147 papers were assessed for eligibility. Of these, only 17 met the criteria for inclusion in this review. To ensure that no citations were missed in this process, the following precautions were taken: (a) citation lists from each database were entered into individual data sheets and cross-checked against each other, and (b) the reference lists of all published studies were checked for relevant citations that may have been missed in a standard search. No additional papers were included as a result of (b). An outline of this process is provided in Figure 1 and complete details of reasons for exclusion are provided in Appendix A.

When planning this review, a specific problem was posed by studies of “non-pathological” dissociative experiences, which can occur in healthy individuals and do not qualify for a psychiatric diagnosis. While some authors consider these experiences to lie on a continuum with dissociative symptoms or disorders [34,35], the majority of contemporary researchers make a clear distinction between “normal” and “pathological” dissociation based on psychometric data [36,37,38]. As only two studies of this phenomenon were included in this review, their findings are discussed along with the rest of the included studies.

The quality of individual studies was assessed using the Q-Genie tool, which is an 11-item scale specifically developed to assess the quality of genetic association studies for the purpose of a literature review or meta-analysis [39]. These guidelines include an item covering the risk of bias, which is reported separately along with the total score for the purposes of this review. For genetic studies containing a comparison group, study quality was rated based on the total Q-Genie score as poor (≤35), moderate (36–45), or good (>45). For studies without a comparison group, the corresponding cut-offs were: poor (≤32), moderate (33–40), and good (>40). Bias was rated on a seven-point scale, with the following categories: poor (1–2), good (3–4), very good (5–6), and excellent (7). Complete scores for each article are provided in Appendix A.

Due to the small number of relevant studies and significant methodological heterogeneity, a formal meta-analysis was not carried out.

## 3. Results

A total of seventeen studies were included in the final review [40,41,42,43,44,45,46,47,48,49,50,51,52,53,54,55,56]. The majority of these were association studies involving a small number of predefined genetic polymorphisms or genotypes, with or without a gene–environment interaction (G×E) component (*n* = 12); there were three genome-wide association studies, one gene expression study, and one study making use of polygenic risk scores. A complete description of these studies is provided in Table 1.

### 3.1. Quality of the Included Studies

Q-Genie scores for each study are provided in the Appendix A. An examination of the total Q-Genie scores for each study found that nine studies were rated as “moderate” in quality” [40,42,43,45,46,48,50,53,54], seven were rated as “good” [44,47,49,51,52,55,56], and only one was rated as “poor” [41]. The mean Q-Genie score was 39.3 for studies without a control group (*n* = 11) and 43.3 for studies with a control group (*n* = 6), indicating an overall “moderate” quality of research in this field.

When evaluating studies specifically in terms of bias, thirteen studies received a rating of “good” and four studies were rated “poor”; no study received a rating of “very good” or “excellent” when evaluated for sources of bias. The mean overall score for bias was 3.1, indicating an overall “good” quality for the included studies.

An additional issue of concern was that several authors, particularly those of association and G×E studies, reported concerns related to study power sample size. Twelve studies received a rating of “poor” (scores 1–2) on this item, and only four studies received a score of “good” or “very good” (scores 3–5). The mean score on this item across all included studies was 2.5, indicating significant concerns related to study power.

### 3.2. Study Populations

None of the included studies focused on patients with a primary diagnosis of dissociative disorder. Two studies were conducted in “normal” adults, with no medical or psychiatric diagnosis, selected from the general population [46,54]. Five studies were conducted in subjects considered to be “high-risk” or vulnerable, including participants from socially and economically deprived backgrounds [47,52] and victims of trauma [41,45,49]. Three studies were based on samples from participants in biobanking programs who had a wide range of medical or psychiatric diagnoses [51,55,56]. The remaining studies were conducted in patients with a primary psychiatric diagnosis other than a dissociative disorder, including obsessive–compulsive and related disorders (OCD) [40,42], borderline personality disorder [44,48], depression [50,53] and bipolar disorder (BD) [43]. The majority of studies were conducted in adults, with only two studies focusing on children and adolescents [41,52]. None of the studies involved a replication of current or past findings in a separate study population.

Ethnicity and gender are important potential confounding factors in studies of neuropsychiatric genetics [57,58,59,60]. Eight of the seventeen studies were conducted in samples where an attempt was made to ensure a degree of ethnic homogeneity: seven studies of subjects of Caucasian/European descent [40,42,45,49,51,55], and one each of Mexican-American [53] and Japanese [54] descent. Three studies recruited only women in an attempt to correct for the influence of gender [47,48,50], while the remainder included both men and women. In all studies with samples of mixed ethnicity or gender, attempts to correct for these confounders were reported in the study methodology and results.

### 3.3. Definition of the Phenotype of Interest

A clear definition of the phenotype being studied in relation to genetic variants is essential in psychiatric genetics, where the categorical diagnoses used in clinical practice may not correspond to meaningful subtypes at the biological level [61]. Only three of the studies included in this review used categorical diagnoses as the phenotype of interest, due to their reliance on medical records, and in these studies dissociative disorders were “lumped” together with anxiety and phobic disorders [51,55,56]. This significantly limits the validity of any conclusions that can be drawn from these studies with regard to dissociation alone. In contrast, the majority of studies [40,41,42,43,44,45,46,47,48,52,54] measured total dissociative symptoms using a standardized rating scale, such as the Dissociative Experiences Scale. Such psychometric instruments provide a total score, a “cut-off” value indicating “pathological dissociation” likely to be of clinical significance, and sub-scale scores for distinct symptom dimensions, such as amnesia and depersonalization/derealization. In theory, this should allow for a more “fine-grained” analysis of dissociative symptoms in relation to genotypes; however, such an analysis was carried out only in a few of the included studies [46,54]. The remaining three studies measured only depersonalization/derealization symptoms and did not examine other sets of symptoms; thus, their findings were applicable only to this specific dimension of dissociative symptomatology [49,50,53]. As noted above, no study including patients with dissociative disorders alone was identified for inclusion in this review. A complete description of the rating scales used in each study is provided in Appendix A.

### 3.4. Associations with Specific Polymorphisms or Genotypes

Twelve of the studies included in this review attempted to establish an association between dissociation and either a single nucleotide polymorphism (SNP) or a genotype involving a limited number of polymorphisms or variants.

Six of these studies focused on genetic variants associated with the monoamine neurotransmitters serotonin, dopamine and noradrenaline, which are associated with stress- and trauma-related disorders [62]. The most commonly studied variant (*n* = 4) was *5-HTTLPR*, a functional polymorphism of the promoter region of the serotonin transporter. Two studies found an association between the “short” (s) allele of this polymorphism, particularly the homozygous s/s genotype, and dissociative symptoms [42,46]; however, two other studies failed to find any association between this polymorphism and dissociation [40,43]. An equally frequent subject of study was the *COMT* gene, encoding the catechol-O-methyltransferase enzyme involved in the catabolism of dopamine and norepinephrine. A study of adults involved in road traffic accidents found an association between a “pain-sensitive” *COMT* haplotype, involving four functional polymorphisms (rs4633-rs4680-rs4818-rs6269 A_C_C_G), and dissociative symptoms [45], while a study of healthy adults found an association between the *COMT* rs4680 (Val158Met) SNP and both total dissociative and depersonalization/derealization symptoms [54]. However, as in the case of *5-HTTLPR*, negative results were obtained in two studies [40,43]. Studies of other monoamine-related polymorphisms or variants, such as those involving the genes for the dopamine type 4 receptor (*DRD4*), the dopamine transporter (*DAT*), the serotonin type 1B and type 2C receptors (*HTR1B, HTR2C*), and the enzymes tryptophan hydroxylase 1 (*TPH1*) and monoamine oxidase A (*MAOA*) did not yield any significant results with reference to dissociation [40,43]

Among other variants studied using an association design, two functional polymorphisms (rs3800373, rs1360870) of the *FKBP5* gene, encoding a protein that regulates glucocorticoid receptor sensitivity [63], were associated with dissociation during and after acute physical injury in children [41]. Possible associations between three functional polymorphisms (rs7607967, rs4371369, rs4387806) of the *SCN9A* gene, encoding a voltage-gated sodium channel expressed in limbic system structures, were observed in women with borderline personality disorder, but these findings were insignificant after statistical correction [44]. The Met allele of the rs6265 polymorphism of the *BDNF* gene, encoding the brain-derived neurotrophic factor, was associated with lower levels of dissociation in patients with bipolar disorder and their relatives [43]; this variant is associated with a reduced susceptibility to post-traumatic stress disorder [64]. A study of the oxytocin receptor gene (*OXTR*), which is associated with stress sensitivity, found an association between homozygosity for the G allele of the rs53576 functional polymorphism in this gene and dissociation, but this was not statistically significant.

### 3.5. Gene–Environment Interactions

There is an increasing amount of evidence that trauma-related disorders arise from a dynamic interaction between genes and the environment [65]. In line with this model, six studies have examined the interaction between a specific genotype and exposure to adverse environmental circumstances in relation to dissociation. Two studies, one of normal adults and one of patients with OCD, found a significant interaction between the *5-HTTLPR* s/s genotype, childhood trauma, and subsequent dissociative symptoms [42,46]; however, a negative result for an s/s x childhood trauma interaction was noted in bipolar patients and their relatives [43]. A study of patients with bipolar disorder and their relatives found a significant interaction between the *COMT* rs4680 Val allele, childhood trauma, and dissociative symptoms [43]. A study of depressed and healthy women found that the rs53576 G/G genotype of *OXTR* was associated with dissociative symptoms in those with a history of unresolved childhood attachment [50]. Finally, two studies examined a haplotype involving four functional polymorphisms (rs3800373, rs9296158, rs1360870, rs9470080 C_A_T_T) of the *FKBP5* gene. While this haplotype was associated with an elevated risk of dissociation in low-income women with a history of childhood trauma [47], it appeared to have the opposite effect in low-income adolescents; adolescents with this haplotype had lower dissociative symptoms even when exposed to significant trauma from an early age [52]. No evidence of gene–environment interaction was observed for the functional variants of the *DRD4* or *DAT* genes in one of these studies [43].

### 3.6. Genome-Wide Association Studies

Three genome-wide association studies were identified in the literature. Only one of these specifically examined dissociative symptoms followed trauma as a phenotype [49]. In this study, no significant associations were found; however, suggestive “peak” associations were reported for the adenylyl cyclase 8 gene (*ADCY8*) and the dipeptidyl-peptidase 6 gene (*DPP6*). The authors did not report any association between post-traumatic dissociation and any of the genes identified through association studies, such as *5HTT*, *COMT* or *FKBP5*, though marginal associations were identified for *FKBP5* and *COMT* which were insignificant after statistical correction. 

The two remaining genome-wide studies, based on large samples from biobanking projects, did not study dissociative disorders or symptoms as a primary objective; they examined a large number of groups of disorders affecting various organs and systems, one of which was “anxiety, phobic and dissociative disorders”. One of these identified a potential association between “anxiety, phobic and dissociative disorders” and a region of chromosome 4 containing the amyloid beta precursor protein-binding family member 2 (*ABPP2*) gene [51]; the other did not report any significant association for this group of disorders [56].

### 3.7. Other Study Designs

Three additional studies on the genetics of dissociative disorders or symptoms did not fit into any specific category and are described here.

In a study of a small sample of women hospitalized with borderline personality disorder, an association was found between dissociative symptoms and the increased expression of the interleukin-6 (*IL6*) gene. Subjects with higher levels of these symptoms also had decreased expression of the following genes: interleukin 1-beta (*IL1B*), mitogen-activated protein kinases-1, -3, and -8 (*MAPK1, MAPK3, MAPK8*), G-protein subunit alpha-I2 (*GNAI2*), arrestin beta-1 and -2 (*ARRB1, ARRB2*), and cyclic AMP responsive element-binding protein 1 (*CREB1*) [48]. 

In a study involving only subjects of Mexican-American ancestry, an analysis of nineteen SNPs identified as associated with depression in earlier research were examined in relation to various symptoms of depression. It was found that this specific set of SNPs was associated with lower depersonalization/derealization symptoms in patients with depression; however, this was not the primary objective of the study [53].

Using polygenic risk scores, a study of 10182 subjects of European descent participating in a biobanking program found that the polygenic risk scores for both depression and bipolar disorder were significantly associated with the group of “anxiety, phobic and dissociative disorders”, suggesting a potential genetic association between these groups of disorders [56].

Specific details of the results of individual studies in terms of scores on standardized rating scales are available in Appendix A.

## 4. Discussion

When compared with other psychiatric disorders or symptom domains, dissociative disorders and symptoms have been relatively under-studied from a genetic perspective. When familial patterns were observed for these disorders, they were often explained exclusively on the basis of psychological mechanisms, such as exposure to trauma or learned behavior [66,67]. Family and twin-based genetic studies of dissociative disorder have suggested that this condition has a substantial heritable component [31,32,68], but the results of these studies were not consistent [69,70] due to methodological limitations and the assumptions made when assessing patterns of inheritance. In contrast with these earlier approaches, molecular genetic studies hold the promise of identifying the biological mechanisms associated with vulnerability to dissociative symptoms and disorders with increasing precision, thus allowing for a more accurate approach to diagnosis and management [22] as well as a deeper understanding of the place of dissociation in psychiatric classification [71,72]. 

In this review, preliminary evidence for an association between dissociative symptomatology and variations in individual genes was identified. The genes implicated are related to monoaminergic transmission (*5-HTT, COMT*), neural plasticity (*BDNF*), neuropeptide receptors (*OXTR*), and the regulation of the hypothalamic–pituitary–adrenal axis (*FKBP5*). These genes have been associated with other stress- and trauma-related symptoms and disorders [73,74,75,76], though results have not always been consistent. Studies in patients with dissociative disorders or symptoms have also found indirect evidence of dysfunction involving these systems. For example, dissociative symptoms are associated with a poorer response to serotonergic antidepressants in patients being treated for other disorders [77,78]; the pharmacological manipulation of serotonergic transmission can induce dissociative symptoms [79,80]; levels of noradrenaline and dopamine are found to be elevated in certain dissociative states [81]; dissociative disorders are associated with higher levels of the neuropeptides oxytocin and prolactin [23]; and pathological and non-pathological dissociative experiences are associated with distinctive alterations in cortisol secretion [82]. While these findings provide a certain degree of biological plausibility to the results of these single-gene association studies, it is also important to note that these findings have not been replicated consistently, and that these loci were identified as being unrelated to, or only marginally associated with, dissociation in a genome-wide analysis [49]. It is therefore unlikely that these variants account for a substantial proportion of the risk for dissociation, though a case could still be made for the role of gene–environment interactions with childhood adversity in the case of the *5-HTT, COMT, OXTR* and *FKBP5* functional polymorphisms.

The results of genome-wide association studies suggest a possible association with loci within the *ADYC8, DPP6,* and *APBB2* genes. There is evidence linking variations in these genes with other psychiatric disorders. *ADCY8* codes for the enzyme adenylyl cyclase, which catalyzes the conversion of adenosine triphosphate into cyclic adenosine monophosphate (cAMP), a key second messenger for several neurotransmitters. It has been associated with comorbid alcohol dependence and depression in women [83], obsessive–compulsive disorder [84], and avoidance behavior related to mood disorders in an animal model [85]. *DPP6*, which encodes a potassium channel subunit related to the excitability of neuronal dendrites and synaptic integration, has been associated with tic disorder [86]. Both *ADYC8* and *DPP6* have also been linked to brain development in childhood [84,87], which may be of relevance to dissociative disorders associated with trauma-related alterations in brain development [88,89]. In contrast, *APBB2*, which encodes an amyloid beta precursor-binding protein, has been associated with dementia [90]. This finding is of equal interest given the emerging evidence of a prospective association between trauma, traumatic stress-related disorders, and subsequent dementia later in life [91]. Despite this supporting evidence, it must be noted that no consistent or highly significant finding has been reported in the GWAS of dissociative symptoms or dissociative disorder; therefore, these results should be interpreted with caution.

The interpretation of other study designs [48,53,55] is less straightforward. However, evidence for increased *IL6* gene expression is consistent with reports of elevated interleukin-6 (IL-6) being associated with dissociative symptoms in depression [92] and with elevated levels of IL-6 in other trauma spectrum disorders [93]. However, IL-6 levels measured immediately after trauma did not appear to predict the course of subsequent trauma-related disorders [94], suggesting that these alterations may appear later in the course of the illness. Likewise, the polygenic association between dissociative disorders and mood disorders is consistent with evidence of high levels of dissociative symptoms in some patients with major depression; some researchers have considered the possibility of a “dissociative depression” subtype associated with childhood abuse and a higher risk of suicide attempts [95], which is consistent with a complex, shared genetic architecture for these disorders.

Several methodological issues should be taken into account when appraising these results. First, though the existing research is of acceptable quality with regard to sources of bias, it is of only “moderate” quality overall, and significant concerns exist with regard to sample size and study power. None of the included studies involved patients with a primary diagnosis of dissociative disorder. Several studies were carried out in patients with another psychiatric diagnosis, introducing a substantial confounding factor. Association studies generally focused on genes related to a limited number of neural or neuroendocrine pathways, many of which are non-specifically associated with a wide range of psychiatric disorders [96,97]. In gene–environment designs, only specific forms of environmental risk, such as childhood abuse by parents or caregivers, were assessed; other relevant forms of traumatic stress, such as bullying by peers, intimate partner violence, or sexual assault after adolescence, were not investigated despite their relevance to dissociation [98,99,100]. In studies measuring dissociative symptoms, there was significant heterogeneity in the instruments used to measure symptom severity, as well as in the specific type(s) of symptoms being studied; in some cases, data on different dimensions of dissociation in relation to the genotype was not analyzed, despite being available. In genome-wide association studies, dissociative disorders were grouped together with anxiety and phobic disorders for analysis in some cases. While this is an inherent limitation of using biobank data in which patient diagnoses are coded using older classificatory systems, it reduces the specificity of any reported findings with reference to dissociation, per se. Not all studies made attempts to correct for confounding factors such as sex and ethnicity. Finally, none of the included studies included a validation of their findings in a separate sample or population. Because of these limitations, it is possible that both false negative (due to low study power) and false positive (due to variations in phenotype, multiple confounding factors, and a lack of replicated associations) results may have been reported in individual studies. In practical terms, this implies that the results reported here are in need of replication before they can truly inform our understanding of the molecular processes underlying dissociative disorders and symptoms.

There are also certain limitations inherent to this review. The retrieval of citations from three specific databases may have led to the inadvertent omission of certain key studies. The marked heterogeneity across studies and the lack of replicated findings for individual associations precluded a formal meta-analysis, limiting the confidence that can be placed in the conclusions drawn above. The current review included only studies of dissociative disorders or symptoms as defined using current criteria; it is possible that conditions such as somatoform and conversion disorders, though placed in separate categories in current systems of classification, may be genetically and neurobiologically linked to dissociation and thus should be studied together [22,101]. As no study included patients with a primary diagnosis of dissociative disorders, as mentioned above, one of the objectives of this review could not be met. Finally, as the review was carried out by a single reviewer, there is a possibility that certain relevant papers may not have been included, though attempts were made to reduce this possibility to the maximum extent possible.

Despite these limitations, the existing evidence suggests the possibility of links between dissociative symptoms and specific genes related to monoamine transmission, neural plasticity, hypothalamic–pituitary–adrenal axis functioning, peptidergic neurotransmission, second messenger signaling, and synaptic integration. Dissociative symptoms may arise from an interaction between functional variants in these genes and early life adversity, particularly childhood abuse. These findings require replication in larger and more homogenous samples, as do the more tentative findings linking dissociation to genes involved in immune functioning. From a clinical perspective, it is possible that the pharmacological manipulation of these pathways may result in the development of better treatments for dissociative disorders. From a more fundamental perspective, there is a need to examine the genetics of dissociation from the point of view of other biomarkers identified in the literature. Potential genetic variants of interest that have not yet been studied include the prolactin receptor [23], opioid peptide receptors [81], genes involved in cytokine signaling [94], and genes involved in lipid metabolism [102]. Studies of gene–environment interactions should move beyond the examination of childhood trauma to include both subtler disruptions in attachment bonds [50] and the effect of traumatic stress later in life, both at an individual level and in survivors of disasters or mass casualties. There is also a need for enhanced genome-wide association studies involving either a more precise definition of the “dissociative” phenotype, or an exploration of shared, “common” genetic factors underlying the co-occurrence of dissociative and other disorders. Epigenetic studies could lead to a more dynamic model of alterations in gene expression and their downstream consequences in patients with clinically significant dissociation [103]. Finally, studies of the genetics of dissociation in relation to other trauma spectrum disorders, such as PTSD and borderline personality disorder, would aid the identification of shared and unique genetic vulnerabilities for this group of disorders. 

## 5. Conclusions

Despite the relatively small number of studies in this area, research on the molecular genetics of dissociative symptoms and disorders has yielded clues pointing both towards conventional, “stress-related” neural mechanisms and novel genetic loci of interest. These results should be interpreted with caution in view of the methodological limitations discussed. However, these findings provide a foundation that can be built upon by further studies with more complex designs, and it is hoped that this will lead to a better understanding of the pathogenesis and treatment of this group of trauma-related disorders.

## Figures and Tables

**Figure 1 genes-13-00843-f001:**
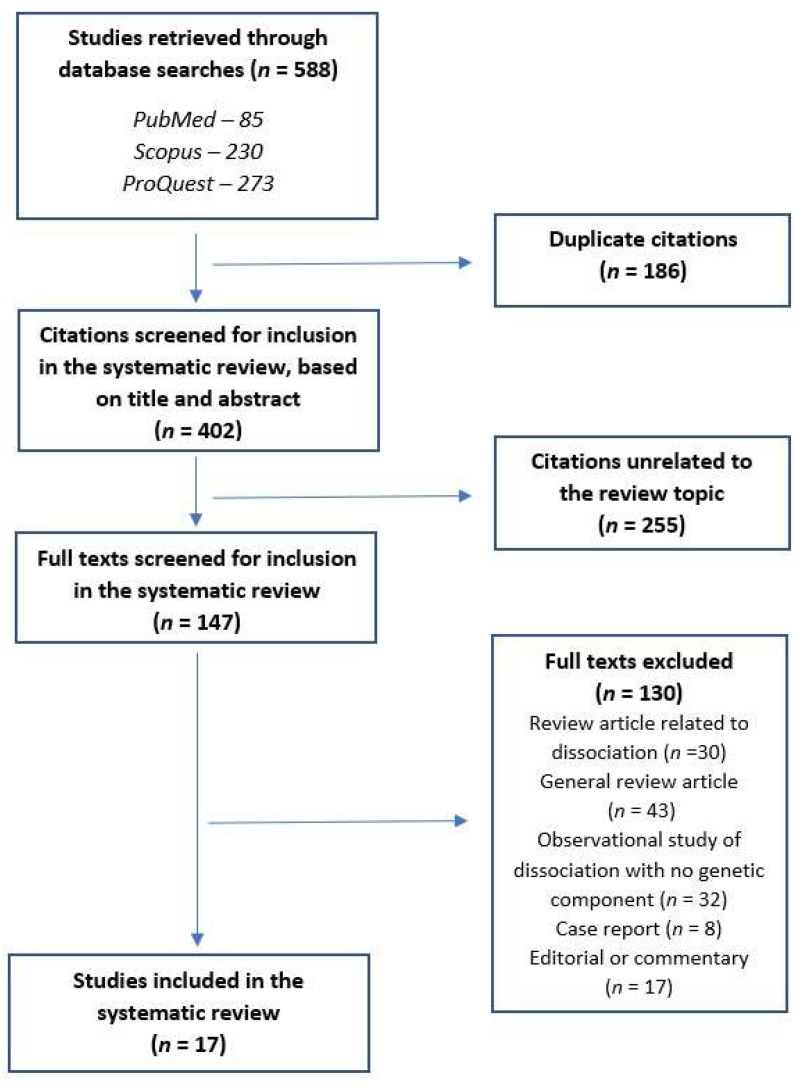
PRISMA flow diagram for the current systematic review.

**Table 1 genes-13-00843-t001:** Summary of studies included in the current review.

Authors	Study Population and Sample Size	Study Design	Phenotype	Polymorphisms Studied, if Applicable	Study Quality-Overall	Study Quality–Bias	Study Results
Lochner et al., 2004 [40]	Patients with obsessive–compulsive disorder or trichotillomania (*n* = 114); Caucasian ethnicity	Association	Dissociative symptoms, total	*DRD4* 48-bp VNTR, *DAT* 40-bp VNTR, *5HTTLPR*, *HTR1B* G861C, *HTR2C* T102C, *TPH1* Val81Met, *COMT* rs4680, *MAOA* C1460T polymorphisms	Moderate	Poor	No association between dissociation and any of the studied polymorphisms.
Koenen et al., 2005 [41]	Children with acute injuries (*n* = 46)	Association	Dissociative symptoms, total	*FKBP5* rs3800373, rs1360870 SNPs	Poor	Poor	*FKBP5* rs3800373 C allele and rs1360870 T allele significantly associated with dissociation during and after injury
Lochner et al., 2007 [42]	Patients with obsessive–compulsive disorder; Caucasian ethnicity (*n* = 83)	Association, G×E (childhood maltreatment)	Dissociative symptoms, total	*5-HTTLPR* polymorphism	Moderate	Good	Interaction between *5-HTT* s (particularly s/s genotype) and childhood trauma associated with dissociation
Savitz et al., 2008 [43]	Patients with bipolar disorder and their relatives (*n* = 178)	Association, G×E (childhood trauma)	Dissociative symptoms, total	*COMT* rs4680, *DRD4* 48-bp VNTR, *BDNF* Val66Met, *5-HTTLPR*, *DAT* 3’-VNTR	Moderate	Poor	*BDNF* Met associated with lower self-reported dissociation; interaction between *COMT* Val and childhood trauma associated with dissociation
Tadic et al., 2009 [44]	Patients with borderline personality disorder (*n* = 161) and healthy controls (*n* = 156); Caucasian ethnicity	Association with control group	Dissociative symptoms, total	*SCN9A* rs16851799, rs7607967, rs4371369, rs4597545, rs4387806, rs6754031, rs12620053, rs13017637, rs12994338, rs4447616 SNPs and haplotype	Good	Good	Possible association between *SCN9A* rs7607967 (G) and dissociation; possible interaction between rs4371369 (G) and rs4387806 (C), sex, and dissociation
McLean et al., 2011 [45]	Motor vehicle accident victims; Caucasian ethnicity (*n* = 89)	Association	Dissociative symptoms, total	*COMT* rs4633, rs4680, rs4818 and rs6269 haplotype	Moderate	Good	Association between “pain-vulnerable” *COMT* haplotype (A_C_C_G) and dissociative symptoms following trauma.
Pieper et al., 2011 [46]	Adult twin pairs (*n* = 184)	Association, G×E (traumatic stress)	Dissociative symptoms, total	*5-HTTLPR* genotype	Moderate	Good	*5-HTTLPR* s/s genotype associated with dissociation in general; s/s genotype associated with pathological dissociation in those with a history of trauma
Dackis et al., 2012 [47]	High-risk, low-income women with (*n* = 170) and without (*n* = 66) childhood maltreatment	G×E (childhood maltreatment) with control group	Dissociative symptoms, total	*FKBP5* rs3800373, rs9296158, rs1360870, rs9470080 haplotype	Good	Good	Interaction between *FKBP5* CATT haplotype and childhood trauma associated with dissociation
Schmahl et al., 2013 [48]	Women hospitalized for borderline personality disorder (*n* = 31)	Gene expression	Dissociative symptoms, total	29 genes selected based on prior associations with depression	Moderate	Good	*IL6* gene expression positively correlated with dissociation; *IL1B, MAPK1, MAPK3, MAPK8, GNAI2, ARRB1, ARRB2, CREB1* expression negatively correlated with dissociation
Wolf et al., 2014 [49]	Adults with a history of trauma exposure (*n* = 484); Caucasian ethnicity	Genome-wide association	Depersonalization / derealization symptoms	Not applicable	Good	Good	No genome-wide significant associations; ten suggestive associations with depersonalization / derealization; highest peaks at *ADCY8* rs263232 and *DPP6* rs71534169; no replication of earlier associations with 5-*HTTLPR, COMT* or *FKBP5*
Reiner et al., 2016 [50]	Pre-menopausal women with depression (*n* = 43) and healthy controls (*n* = 41); Caucasian ethnicity	Association, G×E (unresolved attachment) with control group	Depersonalization / derealization symptoms	*OXTR* rs53576 (A/G) SNP	Moderate	Good	Trend towards higher depersonalization / derealization symptoms in women with the *OXTR* rs53576 GG genotype; interaction between *OXTR* GG genotype and unresolved attachment associated with dissociation.
McCoy et al., 2017 [51]	Patients from academic medical centers participating in biobanking programs (*n* = 10845), Northern European ethnicity	Genome-wide association	Anxiety, phobic and dissociative disorders	Not applicable	Good	Poor	Possible association between the group “anxiety, phobic and dissociative disorders” and locus on chromosome 4 containing the *APBB2* gene.
Yaylaci et al., 2017 [52]	Low-income adolescents with (*n* = 279) and without (*n* = 171) childhood maltreatment	G×E interaction (childhood maltreatment) with control group	Dissociative symptoms, total	FKBP5 rs3800373, rs9296158, rs1360870, rs9470080 haplotype	Good	Good	Possible protective effect of FKBP5 CATT haplotype on dissociation in those with an early-onset and longer duration of maltreatment
Yu et al., 2017 [53]	Patients with depression (*n* = 203) and healthy controls (*n* = 196); Mexican-American ethnicity	Association with control group	Depersonalization/derealization symptoms	19 SNPs identified in a prior study: rs41310573, rs201935337, rs140395831, rs56293203, rs78562453, rs115054458, rs143696449, rs748441912, rs62001028, rs150952348, rs782472239, rs112610420, rs142029931, rs201483250, rs200897153, rs3744550, rs115668237, rs56344012 rs200520741	Moderate	Good	Evidence of a latent depressive subtype associated with 19 SNPs, associated with lower depersonalization / derealization scores
Honma et al., 2018 [54]	Normal individuals (*n* = 76), Japanese ethnicity	Association	Dissociative symptoms, total	COMT rs4680 genotype	Moderate	Good	COMT rs4680 Val/Val genotype associated with total dissociative symptoms and depersonalization/derealization symptoms but not dissociative amnesia symptoms
Kember et al., 2021 [55]	Patients from an academic medical center participating in a biobanking program (*n* = 10182); European ethnicity	Polygenic risk score	Anxiety, phobic and dissociative disorders	PRS for six common psychiatric disorders (schizophrenia, bipolar disorder, depression, attention deficit-hyperactivity disorder and anorexia nervosa)	Good	Good	PRS for depression and bipolar disorder both significantly associated with “anxiety, phobic and dissociative disorders”
Park et al., 2021 [56]	Patients from an academic medical center participating in a biobanking program (*n* = 10,845)	Exome-wide association	Anxiety, phobic and dissociative disorders	Not applicable	Good	Good	No association identified for the group “anxiety, phobic and dissociative disorders”

**Abbreviations**: *5-HTT*, serotonin transporter gene; *5-HTTLPR*, serotonin transporter gene promoter region polymorphism; *ADCY8*, adenylyl cyclase type 8 gene; *APBB2*, amyloid beta-4 precursor protein-binding family B member 2 gene; *ARRB1*, arrestin beta-1 gene; *ARRB2*, arrestin beta-2 gene; *BDNF*, brain-derived neurotrophic factor gene; *COMT*, catechol O-methyltransferase gene; *CREB1*, cyclic AMP response element binding protein 1 gene; *DAT*, dopamine transporter gene; *DPP6*, dipeptidyl peptidase 6 gene; *DRD4*, dopamine type 4 receptor gene; *FKBP5*, FK506 binding protein 5 gene; *GNAI2*, G-protein subunit alpha I2 gene; *HTR1B*, serotonin type 1B receptor gene; *HTR2C*, serotonin type 2C receptor gene; *IL1B*, interleukin-1 beta gene; IL6, interleukin-6 gene; *MAOA*, monoamine oxidase A gene; *MAPK1*, mitogen-activated protein kinase 1 gene; *MAPK3*, mitogen-activated protein kinase 3 gene; *MAPK8*, mitogen-activated protein kinase 8 gene; *OXTR*, oxytocin receptor gene; *PRS*, polygenic risk score; *SCN9A*, sodium voltage-gated channel alpha subunit 9; *SNP*, single nucleotide polymorphism; *TPH1*, tryptophan hydroxylase 1 gene; *VNTR*, variable number of tandem repeats.

## Data Availability

No original data was reported in this study. Data pertinent to the current review is provided in the text, table and Appendix A.

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
