# Peer review of "The Molecular Genetics of Dissociative Symptomatology: A Transdiagnostic Literature Review"

_genes, 2022, doi:10.3390/genes13050843_

Round 1

Reviewer 1 Report

The author presents a systematic review of genetic studies on dissociative symptoms/syndrome studies. A few points to consider:

-Generally systematic reviews have multiple authors enabling them to resolve inclusion "disputes" through consultation. Although I don't think this precludes a single author from performing a systematic review, did you employ any methods to make sure you did not accidentally exclude a study that should have been included (double screening, reference checking, etc)?

-Is it possible to provide any of the standardized dissociation (like the dissociate experiences scale) data in your table or, at least, indicate with a footnote which studies have this data available?

-Can an indicator be added to the table to show which studies had a control group?

-Can you comment on any studies that included validation of their genetic findings in a replication population?

Author Response

I thank the reviewer for their valuable comments on my manuscript. I have made corrections and changes to the paper in accordance with their suggestions to the best of my ability, as follows:

1. Generally systematic reviews have multiple authors enabling them to resolve inclusion "disputes" through consultation. Although I don't think this precludes a single author from performing a systematic review, did you employ any methods to make sure you did not accidentally exclude a study that should have been included (double screening, reference checking, etc)?

Response: I agree with this comment by the reviewer. The following corrections have been added to the paper:

Methodology: "To ensure that no citations were missed in this process, the following precautions were taken: (a) citation lists from each database were entered into individual data sheets and cross-checked against each other, and (b) the reference lists of all published studies were checked for relevant citations that may have been missed in a standard search. No additional papers were included as a result of (b)." (Page 3, lines 103-107)

Discussion - Limitations: "Finally, as the review was carried out by a single reviewer, there is a possibility that certain relevant papers may not have been included, though attempts were made to reduce this possibility to the maximum extent possible." (Page 14, lines 418-421)

2. Is it possible to provide any of the standardized dissociation (like the dissociate experiences scale) data in your table or, at least, indicate with a footnote which studies have this data available? 

Response: I thank the reviewer for this valuable suggestion. A complete list of standardized dissociation data as provided by the original authors has been included in the Supplementary Material as a table which can be consulted for all available details. This table is included in the "Supplementary Material" uploaded with the revised manuscript. This has also been mentioned in the text as follows:

Results: "A complete description of the rating scales used in each study is provided in the Supplementary Material." (Page 9, lines 206-207)

"Specific details of the results of individual studies in terms of scores on standardized rating scales are available in the Supplementary Material." (Page 12, lines 306-307).

3. Can an indicator be added to the table to show which studies had a control group?

Response: I apologize for this omission. In the revised manuscript, the words "with control group" have been added to the "Study Design" column in Table 1 for all relevant studies. Five studies had a control group.

4. Can you comment on any studies that included validation of their genetic findings in a replication population?

Response: I thank the reviewer for this important comment. After checking each of the original papers, it was found that none of the included studies had carried out this important step. This is mentioned in the revised paper as follows:

Methodology: "None of the studies involved a replication of current or past findings in a separate study population." (Page 9, lines 175-176)

Discussion - Limitations: "Finally, none of the included studies included a validation of their findings in a separate sample or population. Because of these limitations, it is possible that both false negative (due to low study power) and false positive (due to variations in phenotype,  multiple confounding factors, and a lack of replicated associations) may have been reported in individual studies." (Page 13, lines 399-404).

Reviewer 2 Report

This is an interesting review focusing on the associations between dissociative symptoms and molecular genetics.

The review has been done carefully according to the PRISMA guidelines, so I do not have any objections to the methodology.

I only have one comment:

The Authors say that: "None of the included studies focused on patients with a primary diagnosis of 160 dissociative disorder". So perhaps you should rethink the title. Maybe a better one would be: "The molecular genetics of dissociative symptoms in different psychiatric disorders: a literature review". Or maybe a different one reflecting the real content of analyzed studied.

Author Response

I thank the reviewer for their valuable comments on my original manuscript. I have made changes and corrections to the paper in accordance with their suggestions to the best of my ability.

1. The Authors say that: "None of the included studies focused on patients with a primary diagnosis of 160 dissociative disorder". So perhaps you should rethink the title. Maybe a better one would be: "The molecular genetics of dissociative symptoms in different psychiatric disorders: a literature review". Or maybe a different one reflecting the real content of analyzed studied. 

Response: I agree completely with this suggestion by the reviewer. The title has been changed accordingly to "The molecular genetics of dissociative symptomatology: a transdiagnostic literature review."